# Real-Time Detection on SPAD Value of Potato Plant Using an In-Field Spectral Imaging Sensor System

**DOI:** 10.3390/s20123430

**Published:** 2020-06-17

**Authors:** Ning Liu, Gang Liu, Hong Sun

**Affiliations:** 1Key Lab of Modern Precision Agriculture System Integration Research, Ministry of Education of China, China Agricultural University, Beijing 100083, China; ningliu@cau.edu.cn (N.L.); pac@cau.edu.cn (G.L.); 2Key Lab of Agricultural Information Acquisition Technology, Ministry of Agricultural of China, China Agricultural University, Beijing 100083, China

**Keywords:** spectral imaging sensor, image segmentation, modified difference vegetation index (MDVI), OTSU, connected domain, potato plants

## Abstract

In this study, a SPAD value detection system was developed based on a 25-wavelength spectral sensor to give a real-time indication of the nutrition distribution of potato plants in the field. Two major advantages of the detection system include the automatic segmentation of spectral images and the real-time detection of SPAD value, a recommended indicating parameter of chlorophyll content. The modified difference vegetation index (MDVI) linking the Otsu algorithm (OTSU) and the connected domain-labeling (CDL) method (MDVI–OTSU–CDL) is proposed to accurately extract the potato plant. Additionally, the segmentation accuracy under different modified coefficients of MDVI was analyzed. Then, the reflectance of potato plants was extracted by the segmented mask images. The partial least squares (PLS) regression was employed to establish the SPAD value detection model based on sensitive variables selected using the uninformative variable elimination (UVE) algorithm. Based on the segmented spectral image and the UVE–PLS model, the visualization distribution map of SPAD value was drawn by pseudo-color processing technology. Finally, the testing dataset was employed to measure the stability and practicality of the developed detection system. This study provides a powerful support for the real-time detection of SPAD value and the distribution of crops in the field.

## 1. Introduction

Potato, having the high nutritional value and yield, is the third largest staple food in the world [1,2]. Chlorophyll, which can capture and transform light energy into chemical energy for the conversion of inorganic matter to organic matter, is an important raw material in potato photosynthesis [3]. In addition, the chlorophyll content is significantly correlated with the content of nitrogen [4], which is one of the most important nutrients for the potato plant’s healthy growth [5], development [6], and production [7]. Therefore, the accurate detection of the chlorophyll content of potato plants is significant for precision agriculture management. In recent years, many destructive and non-destructive methods have been developed to estimate the chlorophyll content of leaves. Destructive methods for detecting chlorophyll content through leaf sampling and laboratory chemical analysis [8]—for example, the spectrophotometer method—are very accurate but time consuming and laborious [9]. These methods cannot rapidly estimate the chlorophyll level of a crop and cannot meet the development of precision agriculture. As a non-destructive and rapid monitoring method, modern spectral analysis technology has shown outstanding advantages for collecting data on a crop’s physical or chemical components, such as the chlorophyll content of plants.

The soil plant analyzer development (SPAD) meter is a rapid, non-destructive spectral device, which is widely employed to measure in situ the chlorophyll content of crops in the field [10,11]. The most commonly used SPAD chlorophyll meter measures transmittance of light through the leaf at 650 and 940 nm to estimate leaf greenness [12]. There are many reports in the literature that the chlorophyll content is high relevant to the SPAD value at leaf scale. For instance, the square of the correlation coefficient (*R*^2^) between SPAD value and the chlorophyll content measured using solvent extraction method in wheat [13], rice [14] and Arabidopsis Thaliana [15] leaves was 0.92, 0.93, 0.93, and 0.98. In addition, the *R*^2^ of 0.95 was found between the chlorophyll content of potato leaves measured in laboratory and SPAD readings [16]. However, the SPAD can only detect the chlorophyll content at a point on of single leaf with low detection efficiency [17,18], it is limited to rapidly estimate the chlorophyll status of whole canopy of crops. Moreover, the SPAD values were different in different measurement spots [19], which is not conducive to the practice of precision agriculture. To measure the spectral data of a canopy area or whole potato plant without any contact or damage, some researchers developed spectral imaging sensors [20,21,22]. For instance, Borhan and Panigrah [21] developed a CCD sensor to collect indoor images at 550 and 700 nm for detecting the SPAD value of potato. And the determination coefficient of calibration set was 0.80. However, the abovementioned study only collected spectral data in the light region, which cannot comprehensively reflect the leaf information to accurately detect the SAPD value. Some studies indicate that the information of some bands located in the near-infrared region is related to SAPD value, such as normalized difference red edge (NDRE _710, 750 nm_**)** index [23], the MERIS terrestrial chlorophyll index (MTCI _681,708, 751 nm_), and red edge positions ranging 702 to 706 nm [24]. On one hand, these results demonstrated the potential to present the SPAD value by high-throughput imaging sensor. On the other hand, they explored the feasibility to estimate the SPAD transmittance of light through the leaf at 650 and 940 nm by the spectral reflectance of canopy within 681–750 nm.

Based on the above problems, in order to obtain the SPAD value and distribution by high-throughput near the ground, we intended to develop a detection system based on a 25-wavelength spectral sensor. The detection system can collect the spectral images of potato plants; process online the spectral data, including image segmentation, reflectance extraction, SPAD value calculation; and draw the visualization distribution map of SPAD value of potato plants in real time. For data processing, the core work in the study consisted of the segmentation of potato plant images and the establishment of a SPAD value detection model. Herein, the segmentation is referred to the process of extracting the target potato plant pixels from spectral images. In the field, the spectral data of potato are disturbed by various noises, such as canopy geometry, soil, weeds, and residual straw, which greatly affect the accuracy of SPAD value detection. Therefore, the accurate segmentation of potato plants from the image is crucial to ensure data validity and establish a high-precision detection model [25]. The Otsu algorithm (OTSU) is widely used for image segmentation, which can automatically obtain the segmentation threshold [26]. However, the OTSU cannot effectively complete the segmentation task if the grayscale values of target object are similar to that of background noises [27]. Combining OTSU with spectral property, many spectral index segmentation methods such as the normalized difference index (NDI-OTSU) [28], excess red index (ExR-OTSU) [29], vegetative index (VEG-OTSU) [30], combined indices (COM-OTSU) [31] have been proposed. These spectral indices have been able to reduce the background information to a certain extent to improve the segmentation performance. However, the segmentation accuracy, which ranged from 53% to 88% for the above methods, was still low [32]. Some learning-based approach segmentation methods have also been proposed, such as a learning approach based on decision trees [33] or on support vector machines [34], the supervised mean-shift method with back propagation neural network [35], or with a Fisher linear discriminant [36] algorithm with high segmentation accuracy for the calibration data. However, these methods rely on a good deal of data to train the segmentation model, and suffer from a long run time, lower stability, and problems with overfitting. Facing complex background noise in the field environment, the modified difference vegetation index (MDVI) along with OTSU and combined with connected domain-labeling (CDL) segmentation method (MDVI–OTSU–CDL) is proposed in this study to extract spectral data of potato plants online. 

The partial least squares (PLS) algorithm is widely used to establish a detection model to describe the relationship between the agronomic parameters and spectral reflectance [37]. The PLS algorithm can reduce the variable collinearity by principal component analysis, which can enhance the model stability [38]. Occasionally, the PLS regression possesses poor prediction accuracy because the spectral variables contain irrelevant information. Some studies have reported that the uninformative variable elimination (UVE), serving as a sensitive wavelength selection algorithm, can improve the performance of the PLS regression model by eliminating invalid variables [39,40]. We attempt herein to establish a high-performance SPAD value model using the UVE–PLS method. 

Therefore, this study aims to develop a detection system based on a novel spectral imaging sensor for the real-time detection of the SPAD value of potato plants in the field. The main data processing steps of the developed sensors system are shown in Figure 1. The system has the following functions:(1)Automatically segment potato plants using the MDVI–OTSU method,(2)Automatically extract the reflectance of potato plants using segmented mask images,(3)Detect the SPAD value of potato plants in real time using the established model,(4)Generate a visualization distribution map of SPAD value of potato plant based on the spectral images and PLS model.

## 2. Materials and Methods

### 2.1. Development of SPAD Value Detection System

In this study, a 25-wavelength spectral imaging sensor (Mode: XIMEAI-5×5-CMOS, Shanghai Branch of IMEC Microelectronics Co., Ltd., Shanghai, China) was used as the hardware part of the detection system, and this sensor is shown in Figure 2a. The filter of this sensor was processed on the wafer of a commercial application CMOS image capture chip that has a mosaic layout. The main features were that there was a specific spectral filter on each pixel, and 25 wavelengths were placed on the COMSIS-CMV2000 sensor with two million pixels. This sensor was able to obtain spectral information of the following 25 wavelengths: 666, 681, 706, 720, 732, 746, 759, 772, 784, 796, 816, 827, 837, 849, 859, 869, 887, 888, 902, 910, 920, 926, 935, 940, and 945 nm. The full width at half maximum (FWHM) varies among different wavelengths, and the FWHM of each wavelength is shown in Table 1. The FWHM gradually widens as the wavelength increase. The sensor had a field of view (FOV) of 50^0^. The image size of each wavelength was 409 × 217 pixels, and the grayscale resolution was 10 bits. Additionally, the sensor is small in size and lightweight, and it supports the secondary development of host computer control software.

In order to realize the real-time detection of SPAD value of potato plants in field, in this research we designed and developed the control software of this spectral imaging sensor using C++ language based on the Qt Creator 4.9.1 platform under a Windows operating environment. The user interface shown in Figure 2b was designed based on the Qt Widgets Application. The image processing functions were realized by calling the OpenCV libraries. The main functions of the software included the following: spectral image collection, exposure time adjustment, spectral image correction, SPAD value pseudo-color expression, SPAD value statistics, and image saving. The operation of the control software was friendly and convenient. 

The spectral sensor and control software comprised the SPAD value real-time detection system. The spectral images of potato plants were collected and the SPAD value was detected via operating the control software after the sensor was connected to the laptop. Before data collection, both the exposure time and focal length need to be adjusted to ensure data validity, and the correction board was collected to correct the reflectance of spectral images.

### 2.2. Data Acquisition

#### 2.2.1. Spectral Image Collection of Potato Plants

The experiment consisted of two parts: The first part, the modeling experiment, was used to design the image segmentation algorithm and to establish the SPAD value detection model. The second part, the testing experiment, was used to test the performance of the developed detection system.

The modeling experiment was conducted from May to June 2019 at the experiment station of China Agricultural University in Haidian district, Beijing, China. The growth stages of potato, as critical phases, included tuber formation stage (S1) and tuber expansion stage (S2). A total of 100 samples were collected using the developed detection system for 50 samples per stage. The testing experiment was conducted on July at the same experiment station using another batch of potato plants. The testing dataset contained a total of 60 samples. The variety of both batches of potato plants was Atlantic. Details about the potato crop growth stages and sampling dates are presented in Table 2. During spectral image collection, the exposure time was set at 50 ms. The imaging sensor was mounted on a stand, and the vertical distance from the sensor to the ground was set at 1.5 m. After collecting one potato plant, the stand with sensor was manually moved the next plant.

#### 2.2.2. SPAD Value Measurement

In order to continuously detect chlorophyll level distribution of whole potato plant in different growth stages, SPAD device (Konica Minolta Co., Ltd., Tokyo, Japan) instead of the laboratory extraction method was employed to measure chlorophyll content of potato plants for establishing the detection model. The SPAD value measurement process is briefly described as follow. Twenty leaves were measured per potato plant sample, and the average SPAD value acted as the true chlorophyll content of this sample. According to the vertical distribution of the leaves on the potato plant, 20 leaves were selected in the order from top to bottom to measure the SPAD value. For each leaf, SPAD value was measured twice, and the measured positions were distributed on both sides of the main vein.

#### 2.2.3. Reflectance Correction of Spectral Images

The reflectance of spectral images was corrected by using Spectralon standard correction board (Labsphere Co., Ltd., North Sutton, NH, USA), in which the four standard gray levels were arranged side by side. The size of the board at each gray level was 6 × 25 cm. The reflectance of the four gray levels of the standard board is 1.0, 0.75, 0.5 and 0 from high to low, respectively. A linear regression method could be used to fit the linear relationship between the grayscale and the reflectance of the four gray levels of the correction board, because both reflectance and grayscale were the parameters that linearly describe the light intensity. The conversion between the reflectance and grayscale of image at a certain wavelength was achieved using Equation (1),
(1)$$R_{\lambda }=a_{\lambda }G_{\lambda }+b_{\lambda }$$
where λ is wavelength, $R_{\lambda }$ is the reflectance at the λ nm, $G_{\lambda }$ is the grayscale at the λ nm, $a_{\lambda }$ and $b_{\lambda }$ are the slope and intercept of the regression equation, respectively. Then, Equation (1) could be used to calculate the reflectance of sample spectral images.

In order to eliminate the influence of light changes of field environmental on spectral data, reflectance correction was performed every 10 min. During each correction, the “Correct” button of the system software was clicked to collect the standard correction board image, and we can manually select the appropriate region from the four gray boards to calculate the four average gray values. The exposure time was set at 50 ms, and the distance from the imaging sensor to the ground was set at 1.5 m. The standard board is collected only during reflectance correction procedure. After calibrating the relationship between gray value and reflectance, the reflectance of sample spectral images will be automatically calculated by using Equation (1). The specific correction method is shown in reference [41], published by our group.

### 2.3. Segmentation Method of Spectral Images

A collected spectral image with a complex background information shown in Figure 3 was selected and analyzed, as it contains both the potato plants (target objects) and background noise information including soil, plastic film, weeds as well as residual straw. Therefore, the accurate segmentation of potato plants need be accomplished to acquire the valid spectral data of the potato plant. Afterwards, the SPAD value detection model was establish based on the spectral data.

#### 2.3.1. MDVI–OTSU Method (Preliminary Segmentation)

The Otsu algorithm (OTSU) proposed by the Japanese scholar Nobuyuki Otsu is an efficient algorithm for image segmentation, which can obtain the segmentation threshold based on the maximum inter-class variance of image grayscale values. However, the OTSU cannot effectively complete the segmentation task if the grayscale values of target object are similar to that of background noises [27]. As is known, different objects possess different spectral properties. Several researchers [29,30,31] have used the difference of reflectance to separate plants from soil, weeds, and other background items. For complex background noises, the modified difference vegetation index (MDVI), along with the OTSU method, was proposed in this study to segment spectral images and extract the potato plant. The MDVI was defined as Equation (2) because the reflectance difference between near-infrared wavelength (888 nm) and red wavelength (681 nm) of potato plants was larger than that of other objects in collected images:(2)$$I_{MDVI(\alpha )}=I_{888}-\alpha I_{681}$$
where $I_{681}$ and $I_{888}$ are the respective spectral image at 888 nm and 681 nm, $I_{MDVI(\alpha )}$ is the calculated image, and α is the modified coefficient, which is 0.5, 1.0, 1.5, 2.0, 2.5, and 3.0, respectively. 

#### 2.3.2. Connected Domain-Labeling Method (Precision Segmentation)

After the preliminary segmentation of the spectral image, noise from items such as soil, mulch, and the correction plate, was removed from the image. However, certain noise information, such as those from weeds and adjacent non-target potato plants, could not be eliminated because their spectral characteristics were similar to that of the potato plants. To solve the above problem, we further used the connected domain-labeling (CDL) segmentation method. The method was implemented based on following steps:

First, the median filtering operation was performed on the binary image obtained by the preliminary segmentation to remove the noise points in the image.

Second, the number of connected domains was counted and the area of each connected domain was calculated in the binary image.

Finally, the largest connected domain was saved as the target potato plant, and other connected domains were removed as noise. Then we obtained the mask image (binary image).

#### 2.3.3. Evaluation of Image Segmentation Accuracy

In this study, a Mean Intersection over Union (MIoU) method was used to verify the accuracy of precision segmentation result. First, an original spectral image shown in Figure 3 was segmented manually by using the Photoshop CS6 software to obtain the reference segmentation image, which is shown in Figure 4. The attributes of the segmented image were binary, which indicated that the grayscale value was 0 (black, background) or 1 (white, target potato). The reference image matrix was denoted as *M*1, and the segmented image matrix was denoted as *M*2. Then, the intersection and union of matrices *M*1 and *M*2 were respectively calculated. Finally, the segmentation accuracy was calculated following the Equation (3):(3)$$A=N_{M1\cap M2}/N_{M1\cup M2}\times 100\%$$
where A is the segmentation accuracy, M1∪M2 and M1∪M2 are, respectively, the intersection and union of *M1* and *M2.*
$N_{M1\cap M2}$ is the number of pixels in intersection image matrix, and $N_{M1\cup M2}$ is the number of pixels in the union image matrix.

### 2.4. Establishment of the Detection Model

#### 2.4.1. Reflectance Spectra Calculation

After the segmentation of potato plants, the spectral reflectance at 25 wavelengths were extracted by mask image for establishing the detection model. The spectral reflectance extraction process is as follows: (1) the corrected spectral images of 25 wavelengths were respectively multiplied by the mask image, then the spectral images of potato plants were obtained. (2) The average reflectance of whole potato plant at 25 wavelengths was calculated. The illustration of spectral reflectance extraction is shown in Figure 5.

#### 2.4.2. Uninformative Variable Elimination

Uninformative variable elimination (UVE) was an algorithm of variable selection based on the variable reliability, which was inspected by the PLS regression coefficient of each variable [39]. It was used to eliminate invalid variables or redundant spectral variables [40]. The PLS model was established using relevant spectral variables and the property variables. The reliability index (RI) was an important indicator to determine whether a variable has informative value.

#### 2.4.3. Partial Least Squares Regression

The partial least squares (PLS) regression proposed by Geladi and Kowalski [42] was used to solve multicollinearity problems among variables. PLS regression simultaneously executes a principal component decomposition on the spectral reflectance matrix and the leaf SPAD value content matrix [43]. The average spectral reflectance of whole potato plant was extracted using the mask image, detailed introduce is shown in Section 2.4.1. The spectral matrix and the SPAD value matrix are correlated in the decomposition process, and a linear regression model is established between them to detect the SPAD value of the potato leaves. Additionally, in order to prevent model overfitting, the internal interaction verification is performed by leave-one-out cross validation, and the optimal latent variation is selected by minimizing the root mean square error of cross validation.

#### 2.4.4. Application of the Dataset

For establishing the detection model, the dataset was divided into a calibration and a validation set according to the ratio of 2:1 by using sample set partitioning based on joint X-Y distance (SPXY) algorithm. According previous report [44], the performance of SPXY was better than the Kennard-Stone (KS) and the random selection (RS) method. The result is presented in Table 3. The calibration set (67 samples) was used to train the PLS model. The validation set (33 samples) was used to verify the established detection model performance. Additionally, the testing set (60 samples) was applied to test the stability and applicability of the developed on-line SPAD value detection system. The performance of the PLS regression model was evaluated with determination coefficient ($R_{v}^{2}$), the root mean square error (RMSE), and the ratio of performance to deviation (RPD), as follows:(4)$$R^{2}=1-\frac{\sum_{i=1}^{n}{\left(y1_{i}-y2_{i}\right)}^{2}}{\sum_{i=1}^{n}{\left(y1_{i}-\bar{y}\right)}^{2}}$$
(5)$$RMSE=\sqrt{\frac{\sum_{i=1}^{n}{\left(y1_{i}-y2_{i}\right)}^{2}}{n}}$$
(6)$$RPD=\frac{Std_{t}}{RMSE_{t}}$$
where $y1_{i}$ and $y2_{i}$ are, respectively, the measured and predicted SPAD value for sample i. $\bar{y}$ is the average value of measured SPAD value. The n is the number of samples applied for calibration, validation or testing set. $Std_{t}$ is the standard derivation of testing set, and $RMSE_{t}$ is the RMSE of the testing set. RPD is an indicator to evaluate the predictive performance of the detection model. The *RPD >* 2, 2 ≥ *RPD* ≥ 1.4, and *RPD* < 1.4 correspond to the model with good, middle, and poor performance, respectively. A scatter plot was created to visually demonstrate the fit and reliability of the detection model.

### 2.5. Visualization Distribution Map of Potato SPAD Value

Each pixel of the spectral image had a spectral reflectance of 25 wavelengths. The reflectance of each pixel was substituted into the established UVE–PLS model to calculate the SPAD value of the corresponding pixel. Then, a grayscale image was obtained, in which the grayscale value represented the SPAD value. Finally, different SPAD value was represented by different colors by calling the Color-map function of OpenCV libraries. The detailed steps are as follows:(1)Extract spectral images of potato leaves at characteristic wavelengths,(2)Extract the reflectance of each pixel in the corresponding characteristic wavelength images,(3)Calculate the SPAD value of each pixel by using the PLSR model to form a grayscale image,(4)Draw the Visualization distribution map of SPAD value of potato by performing pseudo-color processing on the grayscale image.

## 3. Results and Discussion

### 3.1. Spectral Image Segmentation Results

#### 3.1.1. Preliminary Segmentation Results Using the MDVI–OTSU Method

The modified difference vegetation index (MDVI) along with Otsu algorithm (MDVI–OTSU) was applied to eliminate the noise information and to perform the preliminary segmentation of potato plants. In order to highlight the role of MDVI, an image at 887 nm was segmented by using only the Otsu algorithm, and the result is shown in Figure 6. There was a certain removal of the soil background and the standard board at two low gray levels. However, this noise information, including mulch film, weed, residual straw, non-target potato plants and standard correction board at two high gray levels were not eliminated because the potato plant, and these noise objects, had similar grayscale values.

Therefore, the MDVI algorithm was introduced to increase the difference of grayscale values between the target potato plant and the background noises. Only then can the OTSU algorithm fully play the role of image segmentation. The segmentation results using the MDVI–OTSU method under six modified coefficients, 0.5, 1.0, 1.5, 2.0, 2.5, and 3.0, are shown in Figure 7. For MDVI, with the increase of modified coefficient from 0.5 to 3.0, the noises of standard board, soil, weeds, mulch film and residual straws in the image gradually decreased. However, some potato leaves were mistakenly removed as noise, as shown in Figure 7f, when the correction coefficient was too large. Therefore, an appropriate correction coefficient was essential for accurate segmentation of potato plants.

#### 3.1.2. Precision Segmentation Results Using the MDVI–OTSU–CDL Method

After preliminary segmentation, the background noises, except for weeds and adjacent non-target potato plants, had been greatly eliminated, as shown in Figure 7e. The reason was that the weeds, non-target potato, and target potato plant have similar spectral properties. In order to enhance the segmentation accuracy, the segmentation method based on connected domain labeling was applied after preliminary segmentation. The results of precision segmentation are shown in Figure 8. Comparing with the preliminary segmentation (Figure 7), the background noise had been completely eliminated when the modified coefficient was greater than 1.5.

#### 3.1.3. Comparison of Segmentation Accuracy

The accuracy after precision segmentation under six modified coefficients are presented in Table 4. It was found that as the modified coefficient increases in the range from 0.5 to 2.5, the segmentation accuracy gradually improved from 61.99% to 91.41%. Then, the accuracy decreased sharply when the modified coefficient was 3.0 due to excessive segmentation, which was consistent with the analysis results of Figure 8. The 2.5 was selected in the end as the optimal modified coefficient. Therefore, the segmentation strategy with the combined MDVI (2.5)–OTSU method for preliminary segmentation with the CDL method for precision segmentation was applied to extract potato plants with satisfactory results. Additionally, the optimal segmentation strategy was denoted as MDVI (2.5)–OTSU–CDL. The segmentation performance of different modified coefficients was further analyzed by model results in Section 3.3.1.

### 3.2. Data Characteristics Analysis

#### 3.2.1. Spectral Response of Potato Plant at 25 Wavelengths

In order to analyze the spectral characteristics of different objects, we manually selected the region of interest of each object. The average reflectance at 25 wavelengths of the target potato plant, soil, weeds and mulch film was manually extracted from the one of corrected spectral images, as shown in Figure 9. Potato plants had low reflectance in the red wavelength due to the strong absorption of the leaf pigment and had the high reflectance in near-infrared wavelengths due to the strong reflection of the cell cavity in leaves. Weeds possessed the similar spectral responses because weeds are also green plants. The mulch film exhibited high reflectance in the range of 660–950 nm, because it was made of polyethylene that had no characteristic absorption in visible and near-infrared wavelengths. However, the reflectance of soil was very low at 25 wavelengths due to the spectral absorption of complex components in the soil.

#### 3.2.2. SPAD Value Statistics of Modeling Dataset 

Figure 10 shows the SPAD values at the tuber formation (S1) and tuber expansion (S2) stage. The average SPAD value of potato was 41.99 at S1 and 32.22 at S2, respectively. It was found the SPAD value of S1 was higher than that of S2. The reason was that the stems and leaves of the potato plant grew vigorously at the tuber formation stage, after which the potato was dominated by tuber growth and the leaves gradually withered.

### 3.3. Detection of SPAD Value of Potato Plants

#### 3.3.1. Influence of Modified Coefficients on PLS Model

In order to compare the above six modified coefficients on SPAD value detection model accuracy, the average reflectance at 25 wavelengths of each potato plant were extracted automatically by using the mask images obtained from six segmentation methods. Then, the six PLS models were established respectively based on their 25 wavelength reflectance. The model results are shown in Table 5. The determination coefficient ($R_{v}^{2}$) and root mean square error (RSMEV) of validation set were used as the indicators to further evaluate the performance of segmentation methods. As can be seen from Table 5, with the increase of the modified coefficient from 0.5 to 2.5, the $R_{v}^{2}$ value gradually increased from 0.658 to 0.822, and the RMSEV gradually decreased from 5.242 to 3.810. The reason was that the noise information was gradually eliminated with the increase of the α value. However, the PLS performance sharp decreased when α was 3.0, which was due to the fact that some spectral data of potato plant were mistakenly eliminated as noise. Therefore, the MDVI (2.5)–OTSU–CDL was the effective method to segment the potato plant. The PLS model results were consistent with the image segmentation results (in Section 3.1.3). The subsequent modeling analysis applied the spectral data extracted by MDVI (2.5)–OTSU–CDL method.

#### 3.3.2. Sensitive Variables Selection

The average reflectance at 25 wavelengths of each potato plant were extracted automatically by using the mask image obtained using the MDVI (2.5)–OTSU–CDL method. Therefore, spectral data of each potato sample consisted of the reflectance of 25 wavelengths. To ensure the stability of the PLS detection model, the sensitive variables of SPAD value were selected by using UVE. The UVE result is shown in Figure 11. The variables with higher reliability index contribute more to the detection model. With a threshold of 1.0, 10 wavelengths were selected as the sensitive variable, and they were 681, 706, 816, 837, 849, 859, 869, 888, 910, and 935 nm. 

#### 3.3.3. Establishment of UVE–PLS Model

The UVE–PLS model was established based on these sensitive variables. To highlight the role of UVE, the PLS detection model also was established based on 25 wavelength variables. The model results are presented in Table 6. In the modeling, both PLS and UVE–PLS adopted the leave-one-out cross validation for internal interactive validation to select the optimal number of principal components (PCs). The results are shown in Table 6. For the PLS model, the number of PCs was 8. For UVE PLS model, the number of PCs was 6.

The determination coefficient of calibration set ($R_{c}^{2}$) of PLS (0.887) was higher than that of UVE–PLS (0.864) because the PLS model possessed more input variables. However, The RPD of UVE–PLS (2.461) model was higher than that of PLS (2.014), which demonstrated that the UVE–PLS had good stability. The *RMSEV* of UVE–PLS (3.315) was lower than that of PLS (3.810), which revealed that the former had excellent predictive accuracy. The prediction results of PLS and UVE–PLS model are shown in Figure 12. Compared with the prediction results of PLS (Figure 12a), the predicted values of validation set of UVE–PLS were evenly distributed in both sides of the 1:1 line, which further illustrated the performance of UVE–PLS model was better for the SPAD value detection of potato plants.

For UVE–PLS model, the linear regression between the reflectance of sensitive wavelengths and the SPAD value is shown as Equation (7):(7)$$\begin{array}{l} Y=-8.651R_{681}+1.287R_{816}-0.687R_{827}+0.251R_{837}+0.089R_{849}\\ -2.010R_{859}+1.977R_{869}-0.795R_{888}+0.366R_{910}+6.190R_{935}+43.828 \end{array}$$
where Y is the predictive values of SPAD value and R is the reflectance of each sensitive wavelength. 

### 3.4. Visualization Distribution Map of SPAD Value

Two potato plant samples were randomly selected from the tuber formation stage and tuber expansion stage, respectively. The pseudo-color visualization distribution map of SPAD value of two potato plants was drawn according to that explained in Section 2.5. The result is shown in Figure 13. The average SPAD values of the potato plants were 41.425 at S1 and 27.512 at S2. In the pseudo-color map, different SPAD values at each pixel were represented by different colors, and the color ranged from blue to yellow (blue–green–yellow), which in turn indicated that the SPAD value was from low to high. The background color was blue, which indicated that the SPAD value was 0.

The following conclusions were obtained by analyzing Figure 13. The SPAD value of the potato plant at S1 was higher than that at S2, which was consistent with the SPAD value statistics in Section 3.2.2. In addition, it was found that the SPAD value was higher in the center of the potato plant and was lower in the edge. The reason might be that the nutrient substance was transported from the central main stems to the branches. From the perspective of image collection, the central position of the pseudo-color map was the upper part of potato plant, the upper leaves were nascent leaves, which had more active photosynthesis. The edge of the map was the lower part of potato plant, the lower leaves were more withered. This result was consistent with the conclusion obtained by that of a previous study [20], which indicated the SPAD value of potato plants leaves gradually decreased from the upper part to the lower part.

### 3.5. Testing of the Developed Detection System

In order to test the performance of developed SPAD value detection system such as the stability and applicability, the MDVI (2.5)–OTSU–CDL method was employed to segment the potato plant and further extract the spectral data of testing dataset (60 samples), and the spectral data were substituted into the UVE–PLS detection model to predict SPAD value. The scatter plot of 1:1 was created, as shown in Figure 14, to visually demonstrate the SPAD value prediction results. The UVE–PLS model showed a strong *R*^2^ value of 0.776 and a small RMSE value of 3.621, which illustrated that the developed system possessed good detection capability. These SPAD value values were evenly distributed on both sides of the 1:1 line, further illustrating that the developed system possessed good stability. 

## 4. Conclusions

This study developed a SPAD value real-time detection system based on a spectral imaging sensor. The detection system had two core functions, which were potato plant segmentation and SPAD value detection. To accurately extract the potato plant from the spectral images, the MDVI–OTSU–CDL segmentation method was proposed, and the six modified coefficients were discussed. The results showed that the segmentation accuracy was best when the modified coefficient was 2.5 (α=2.5), so the MDVI (2.5)–OTSU–CDL was applied to segment the potato plant. Then, the reflectance at 25 wavelengths was extracted by the segmented mask images. The UVE–PLS model was established to detect accurately the SPAD value of potato plants, and the $R_{v}^{2}$, RMSEV, and RPD of the UVE–PLS were 0.850, 3.31, 2.461, respectively, which resulted from the sensitive variables containing the work wavelengths of SPAD meter. The pseudo-colored map could visualize the distribution of SPAD value. Finally, the performance of the developed detection system was measured using the testing dataset; the *R*^2^, RMSE, and RPD were 0.776, 2.478, and 1.891, respectively, which demonstrated that the developed detection system possessed good stability and excellent applicability.

There were seven sensitive wavelengths selected by UVE, the number of which was more than the SPAD meter (two wavelengths) but less than the PLS model. The model results showed that the UVE–PLS possessed better prediction ability, which resulted from the UVE having eliminated the uninformative variables. The differences of sensitive wavelengths may be due to that the developed sensor system collects the reflectance data and the SPAD meter collects the transmittance data. SPAD is a non-destructive device to measure the chlorophyll content and its value has a strong correlation with the leaf chlorophyll of the potato plants. In the paper, to continuously detect chlorophyll level distribution of whole potato plant in different growth stages, the SPAD device instead of the laboratory extraction method was employed to measure chlorophyll level of potato plants for establishing the detection model. The distribution of SPAD can be obtained using the sensor system, but the relationship between SPAD value and chlorophyll content need to be calibrated, and then the results can represent chlorophyll content.

Overall, the new proposed system is helpful for detection chlorophyll level of potato plants. However, the method in this study is based on specific spectral data for potato crops. The restrictions are based on the existence of other datasets or potato varieties. Therefore, more datasets from a wide range of potato varieties, planting patterns, and experimental fields should be collected and analyzed to improve the detection performance of the developed sensor system. In addition, this proposed system can be employed to detect the chlorophyll content of others crops after improving the segmentation method and the detection model and adjusting the imaging setups.

## Figures and Tables

**Figure 1 sensors-20-03430-f001:**
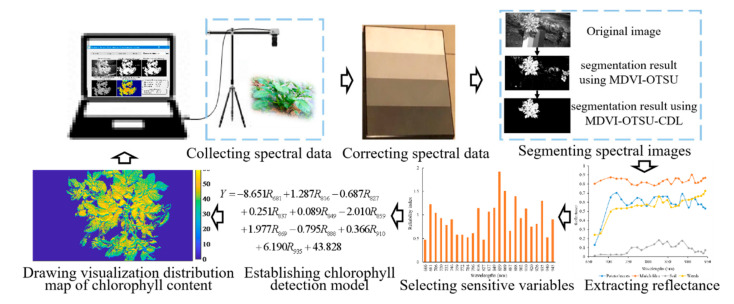
Schematic illustration of the main data processing steps.

**Figure 2 sensors-20-03430-f002:**
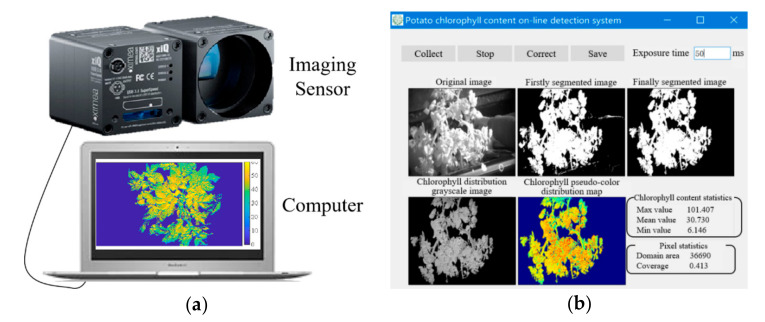
Spectral sensor and the control software of detection system. (**a**) Spectral sensor; (**b**) Control software interface.

**Figure 3 sensors-20-03430-f003:**
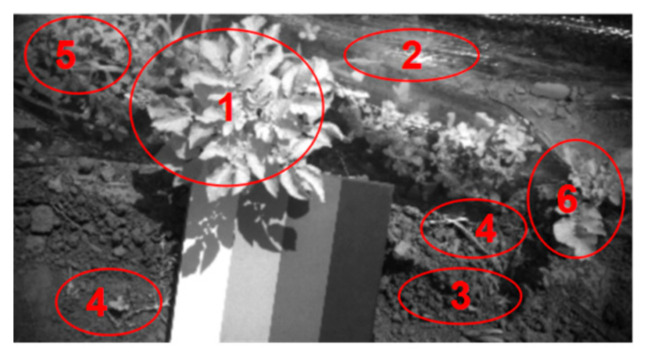
Image of a potato plant at 888 nm in the field. **1**: target plant, **2**: mulch film, **3**: soil, **4**: residual straw; **5**: weeds, **6**: non-target potato plant.

**Figure 4 sensors-20-03430-f004:**
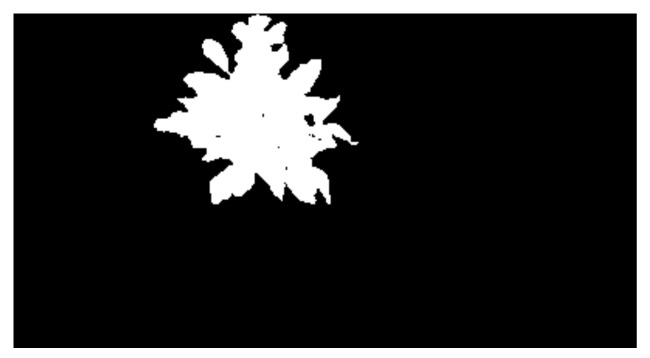
The reference segmentation image.

**Figure 5 sensors-20-03430-f005:**
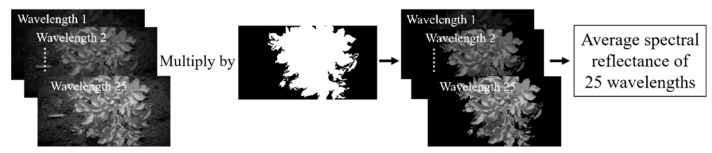
The spectral reflectance extraction process.

**Figure 6 sensors-20-03430-f006:**
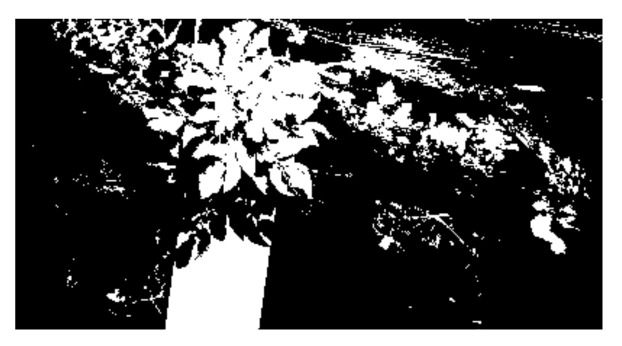
Segmentation result using OTSU algorithm of the image at 887 nm.

**Figure 7 sensors-20-03430-f007:**
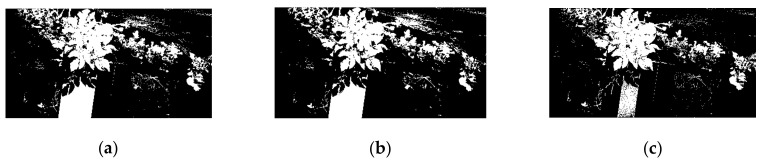
Segmentation result using MDVI–OTSU method under six modified coefficients. (**a**) α=0.5; (**b**) α=1.0; (**c**) α=1.5; (**d**) α=2.0; (**e**) α=2.5; (**f**) α=3.0.

**Figure 8 sensors-20-03430-f008:**
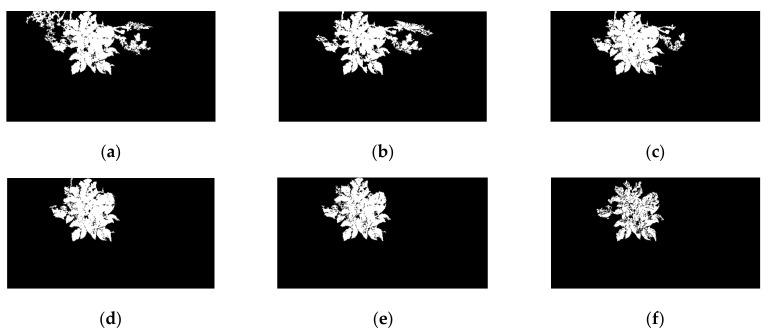
Precision segmentation result after MDVI–OTSU under six modified coefficients. (**a**) α=0.5; (**b**) α=1.0; (**c**) α=1.5; (**d**) α=2.0; (**e**) α=2.5; (**f**) α=3.0.

**Figure 9 sensors-20-03430-f009:**
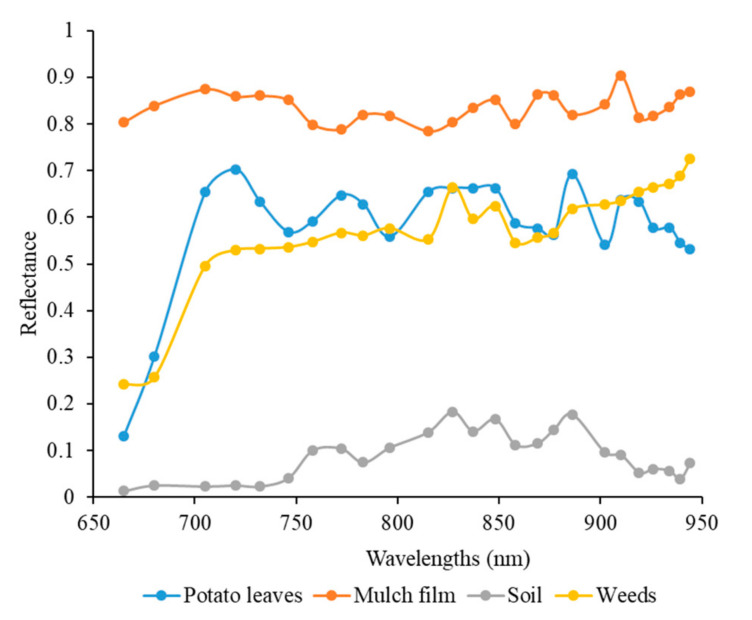
Reflectance of potato leaves, mulch film, soil and weeds at 25 wavelengths.

**Figure 10 sensors-20-03430-f010:**
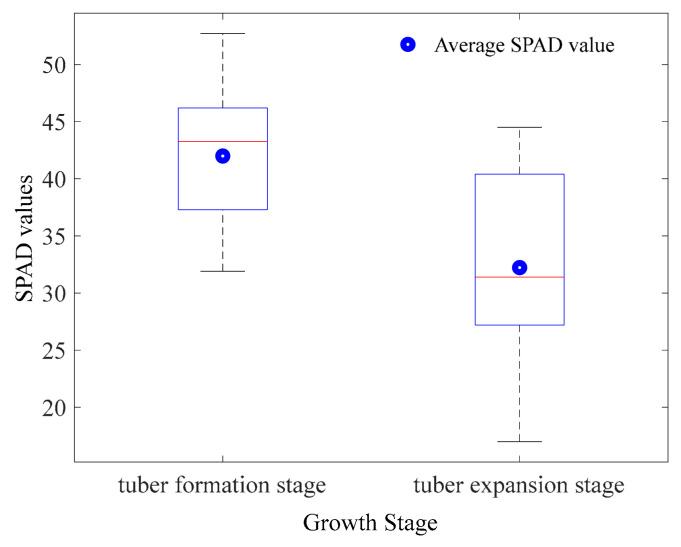
The boxplot of the SPAD value statistics of potato plants at two stages.

**Figure 11 sensors-20-03430-f011:**
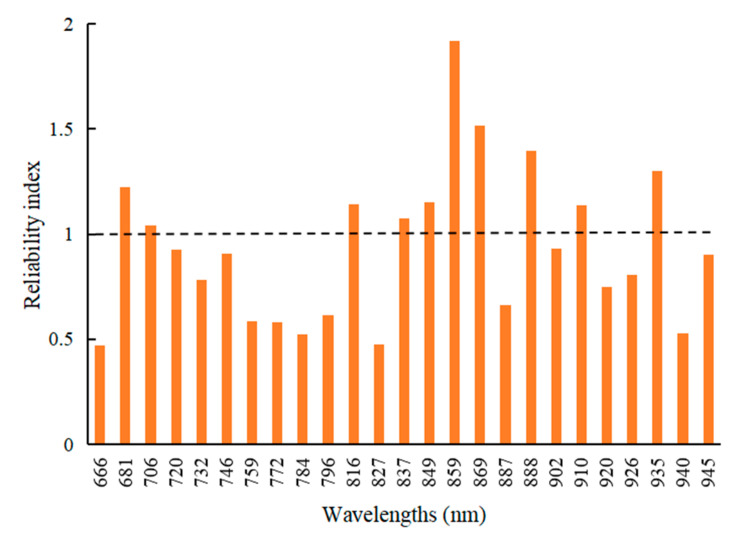
The run result of uninformative variable elimination.

**Figure 12 sensors-20-03430-f012:**
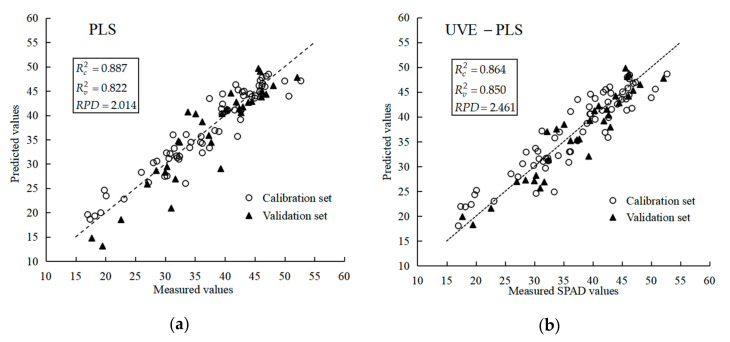
The prediction result of PLS and UVE–PLS models. (**a**) PLS model; (**b**) UVE–PLS model.

**Figure 13 sensors-20-03430-f013:**
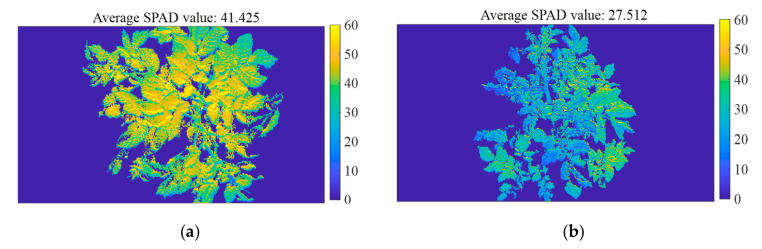
The visualization distribution map of potato plant at two growth stages. (**a**) tuber formation stage; (**b**) tuber expansion stage.

**Figure 14 sensors-20-03430-f014:**
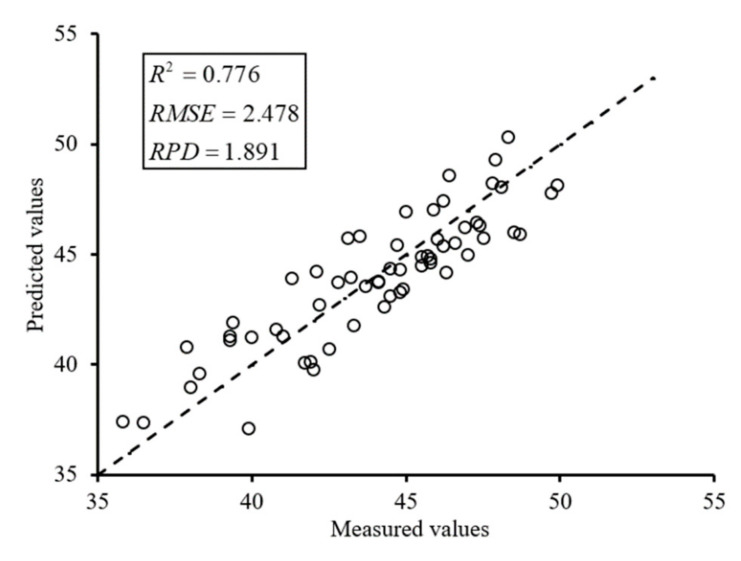
Testing result of developed detection system.

**Table 1 sensors-20-03430-t001:** The FWHM values of different wavelengths.

Wavelength (nm)	FWHM (nm)
666, 681, 706, 720, 732, 746, 759	3.35
772, 784, 796, 816, 827	4.69
837, 849, 859, 869, 887	6.04
888, 902, 910, 920, 926	7.39
935, 940, 945	12.10

**Table 2 sensors-20-03430-t002:** Details of the spectral data collection.

Experiment Part	Growth Stage	Collection Date	Sample Number
Modeling experiment	S1	24 May	50
	S2	15 June	50
Testing experiment	S1	18 July	30
	S2	27 July	30

**Table 3 sensors-20-03430-t003:** SPAD values 1 statistics of calibration set and validation set.

Dataset	Samples	Maximum	Minimum	Average	STD ^2^
Calibration	67	52.70	17.00	36.13	8.77
Validation	33	50.00	17.70	39.09	8.14
Testing set	60	50.90	35.8	44.04	3.32

* SPAD refers to the soil plants analyzer device; ^2^ STD refers to the standard deviation.

**Table 4 sensors-20-03430-t004:** Statistics of segmentation accuracy under different modified coefficients (α ).

α	0.5	1.0	1.5	2.0	2.5	3.0
Accuracy	59.19%	70.08%	83.58%	88.08%	**91.41%**	77.24%

**Table 5 sensors-20-03430-t005:** Validation set results of PLS models of six segmentation methods.

Segmentation Methods	α	$R_{v}^{2}$	RMSEV
MDVI(0.5)–OTSU–CDL	0.5	0.658	5.242
MDVI(1.0)–OTSU–CDL	1.0	0.678	5.053
MDVI(1.5)–OTSU–CDL	1.5	0.690	5.021
MDVI(2.0)–OTSU–CDL	2.0	0.781	4.006
MDVI(2.5)–OTSU–CDL	2.5	0.822	3.810
MDVI(3.0)–OTSU–CDL	3.0	0.735	4.468

**Table 6 sensors-20-03430-t006:** Statistical results of SPAD value detection models.

Model	Inputs	PCs	Calibration Set		Validation Set		RPD
			$R_{{}_{c}}^{2}$	RMSEC	$R_{{}_{v}}^{2}$	RMSEV	
PLS	25	8	0.887	2.830	0.822	3.810	2.014
UVE–PLS	10	6	0.864	3.208	0.850	3.315	2.461

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
