# Peer review of "Real-Time Detection on SPAD Value of Potato Plant Using an In-Field Spectral Imaging Sensor System"

_sensors, 2020, doi:10.3390/s20123430_

Round 1
Reviewer 1 Report
Review of the paper “Real-time detection of potato plant chlorophyll content by an in-field spectral imaging sensor system”. Sensors.
General comments:
The authors propose an instrument to estimate the chlorophyll content of potato plants. The main defect of the paper is that the instrument does not directly measure chlorophyll, as it is repeatedly stated in the article, and even in the title. It has been calibrated with SPAD, which is an instrument that provides measurements in digital counts (DC), which can be related to the chlorophyll content. Many articles by different authors study the calibration of SPAD in different plant species, showing that these calibrations can change depending on the type of crop. In summary, the instrument provides measurements that correlate with SPAD DC, which in turn correlates with chlorophyll, but no direct measurement of chlorophyll content is given. To really measure chlorophyll content, the instrument should be calibrated with direct measurements of leaves chlorophyll (e.g. with laboratory extraction).
On the other hand, chlorophyll absorbs light in the red and the blue regions of the visible spectrum. From 700 or 750 nm, no light is absorbed. Therefore, it is not understood that an instrument to measure chlorophyll uses bands between 666 and 950 nm, losing also the green region, where chlorophyll mainly reflects.
The English of the whole article should be improved.
The article should be kept to the template of the journal, such as removing spaces between paragraphs, font size of tables, etc.
The methodology followed in the field should be more specific, providing details of the protocols followed, where the leaves on which the data were taken with the SPAD instrument were located, etc.
Specific comments:
Lines 2, 19, 22, 66, 106, 274, 341 and others… is not chlorophyll content, these are SPAD DC
Line 56, Wu [18]… >>> Qian et al. [18]…
Line 58, Borhan [19] >>> Borhan and Panigrah [19]…
Line 120, FOV of 50º
Line 160, specify how the measurement leaves are distributed
Line 181, do not repeat “image”
Line 235, Geladi [36] >> Geladi and Kowalski [36]
Line 237, spectral reflectance matrix >>> explain better how the spectrum is obtained. From a single pixel? Mean multiple pixels? Mean one leaf?
Line 335, Figure 9. Shows >>> Figure 9 shows
Line 374, was >>> were
Line 364, “The determination coefficient of validation set ( R2 ) of UVE-PLS (0.850) was higher than that of PLS (0.822), which revealed that the former had excellent predictive accuracy” is not true, R2 may be high but the predictive accuracy of the method is given by the RMSE
Author Response
Please see the attachment "reply for reviewer 1".

Reviewer 2 Report
The premise of the paper seems solid enough. The methods used are quite standard and well known, but their combination for potato chlorophyll detection seems novel. There are however some major issues that should be addressed (most notably the issues with the segmentation evaluation):
- The reflectance calculation is not very clearly explained, and seems insufficient. Figure 1 seems to imply that the imaging was done from directly above the plant, but this is not explained in detail. The impact of the reflectance calibration for the results should be considered and discussed, even if the goal is to eventually use a drone with a different calibration approach. G There is also no mention of how the grey values were picked from the images, or how the correction was applied to the whole image. This could affect the conclusions, especially about the variation seen in figure 11 which could be due to variation in the light, not the plant.
- The definition of the MDVI in (eq 2, line 195) is missing the parameter alpha from the right side of the equation, making it impossible to know what the actual effect of the parameter is, and no references are given to any other uses of this specific index, if they exist. Since the alpha is chosen a specific value for the segmentation this should really be fixed, and some discussion of the role of alpha in the MDVI and it's effect on the segmentation would be welcome.
- Given table 3, it seems that the evaluation of the segmentation accuracy is only performed for the whole dataset and thus does not indicate generalizability, which is suprising given that you could have used the same split into calibration and training datasets that was used elsewhere for this evaluation as well. Also, the accuracy metric (Eq 3) only determines the rate of true positives (pixels identified as potato both manually and automatically), and for a proper evaluation it would be better to compute the full confusion matrix (true/false positives/negatives). This is important to show, since even a segmentation method that flagged the whole image white would show as 100% accurate by your metric alone.
- The need to have the target plant as the biggest object in the image for the segmentation to work is a big constraint, and is not really possible to achieve in a drone setting, for instance. Since SPAD is used as the reference method, it would be good to give some more motivation for and information about the intended use of the device and the imaging procedure, since it seems that given the need for calibration targets and other imaging constraints it would seem simpler to just use the SPAD device directly in this specific setting.
Some more minor issues in the text, per line numbers:
29: reference 1 does not seem relevant to the sentence
30-32: unnecessary repetition of chlorophyll importance
32: it would seem more proper to cite the source in [3] (Muños Huerta et. al 2013) directly for this.
45-46: "analytical spectral devices" and "hyperspectral sorting instruments" are odd choices of words (translation issues?), and the whole paragraph doesn't really describe the referenced methods properly.
61-63: it is not the number of wavelengths that is the issue, but the width of the bands (the SPAD only uses two wavelengths, and it's your reference!)
96: "noise" rather than "invalid information"
109: The quality and size of figure 1 makes the text hard to read
119: What are the widths (FWHM) of the spectral bands?
120: "field of view of 50" is missing units
151: saying "medium fertility" is useless without a reference
152: extra "all" in sentence
195: Fix the equation, alpha missing from right side
241: ...selected by minimizing the root mean square error...
246: As written, this seems to imply that you tried different partitionings and picked the one with the best performance, which I sure hope wasn't the case. You should make it clear what performance was better.
247: The reference doesn't exist (which it really should, to clear the previous confusion)
254-256: bad typesetting of the equations make them hard to read (especially y and y')
257: sentence beginning with And
260: Bad typesetting of the inequalities
264: SPADrefers in the caption missing space
319: DCL instead of CDL
323: Were these single-pixel spectra, or averages of some area?
327: Weeds being green plants is not really relevant here, since the wavelengths in the study are mostly red and near-infrared (and it would be a tautology otherwise).
344: extract automatically
348: Why was 1.0 chosen as the threshold?
352: figure 10, marking a horizontal line at the threshold value would be helpful to the reader
360-361: table 4, why is the RMSEC of the PLS smaller on the calibration set? You could show the scatterplots.
367: Is the difference in R2 really meaningful? Again, plots might tell more.
382: figure 11, the blue background does not seem to be zero as compared to the color map, you should check the accuracy of the images. Also, you should consider not using a rainbow colormap for linear values, as they are known to mislead the eye in some cases
417: UVE having eliminated
Author Response
Please see the attachment "reply for reviewer 2".

Round 2
Reviewer 1 Report
The article has been improved and the authors have responded successfully to the questions and suggestions. The paper can be published.
Author Response
Thank you for your approval.
Reviewer 2 Report
The issue of missing control for the segmentation method is not answered. If you select the parameter using a single image, there is no guarantee that it is the best choice for the rest of them. Since you use connected domain labeling, the segmentation result is image-wide and not pixel-specific, and as such each pixel can't be considered a separate experiment (which would be the case if you compared the preliminary segmentation with your ground truth). If you do not want to manually segment more images to confirm the method (which I understand), you could justify your choice by selecting alpha based on the preliminary segmentation and running the full procedure (segmentation + UVE-PLS) separately with and without the CDL to get better statistics (or also run the whole thing with each alpha).
The new metric (MIoU) for the segmentation is better, though it would still be good to also provide the other per-error metrics (N_(M1 \ M2) and N_(M2 \ M1) as proportion of N_(M1 U M2)), even if you use MIoU as the single number for deciding alpha.
There is still very little description of how the reference grey values are extracted from the photos for the reflectance correction, and no information how the imaging procedure was carried out in the field:
- Was a reference plaque placed in each image, or only in some of them?
- How were the reference reflectance values extracted? By hand or automatically by software, taking a mean value or a single pixel?
- Your description of the correction being performed every 10 minutes suggests that this procedure was carried out in the imaging software during imaging (instead of being computed afterwards during postprocessing as is often the case), but this is not clear. It would be better if you described the physical imaging process in the field.
There are references to direct chlorophyll detection (instead of SPAD value which you changed most to) in many places in the paper (111-115, 134-146, 363-364, 397 and probably others I missed)
101: I agree with your comment that "noise" might be confusing in this context. "irrelevant" might still be better than "invalid".
124-127: A table of wavelengths and FWHM values might be clearer to read.
281 and 285: The bar above y is still badly typeset (too far).
284: Extraneous "And"
352: Extraneous "And then"
404: "predicted" should be "prediction"
448: "contained" should be "containing". You should also discuss this a bit more, since it is not obvious why you couldn't just use the SPAD wavelengths (the difference between reflectance and transmittance measurements)
454: "needs" should be "need"
Round 3
Reviewer 2 Report
The results in the paper do indicate that SPAD measurements can be modeled to an extent using reference measurements and reflectance data, and that using UVE can slightly improve the accuracy. The segmentation method used places some constraints on the imaging setup, but seems to be sufficient if not very easily generalizable. The presented modified difference vegetation index could be of larger interest as it seemed to be somewhat able to discriminate between weeds and planted potatoes. The methods used are as such not novel, but the specific presented application seems useful and worthy of study.
The measurement procedure could still be explained in more detail, with regard to the physical setup used for the imaging. The images do suggest a stand that presumably was moved manually from plant to plant, but this is not explained in the text.
The conclusions section in the paper could be improved. There is next to no discussion on how the results generalize to different plants, ranges of SPAD values outside the tested potatoes, or to different imaging setups. There should also be some discussion on the scale of the remaining prediction error in the method and how it relates to the intended use of the setup. It would generally be good for the authors to think about and discuss the limitations of the study, even if it is a preliminary one.
The paper also still has some minor issues with English grammar and style in parts, but is easily understandable.
116: If this sensor is the same as the one in reference 41, it would be good to reference it here as well.
123-127: No need to repeat the values that are in the table, just refer to it.
189: bad grammar, change to: "The "Correct" button of the system software was clicked to..."
192: Unnecessary "Also"
193: Extra "And"
289: "...SPAD value. And n..." should either have a comma, or no "And".
346-349: You should reference and compare this with the results in 3.3.1
378: Extra "And"
412: Extra "And"
421: The figure caption should have "prediction" instead of "predicted"
465: The non-UVE PLS also included the work wavelengths and it had worse performance, so I think you are trying to say something else here.
Author Response
Please see the attachment, thank you.
